# Two-epoch cross-sectional case record review protocol comparing quality of care of hospital emergency admissions at weekends versus weekdays

Julian Bion,[1] Cassie P Aldridge,[1] Alan Girling,[2] Gavin Rudge,[2] Chris Beet,[3] Tim Evans,[4] R Mark Temple,[5] Chris Roseveare,[6] Mike Clancy,[7] Amunpreet Boyal,[1] Carolyn Tarrant,[8] Elizabeth Sutton,[8] Jianxia Sun,[1] Peter Rees,[9] Russell Mannion,[10] Yen-Fu Chen,[11] Samuel Ian Watson,[11] Richard Lilford,[11] On behalf of the HiSLAC collaboration

For numbered affiliations see end of article.

**Correspondence to**
Professor Julian Bion;
c.s.price@bham.ac.uk

## ABSTRACT

**Introduction** The mortality associated with weekend admission to hospital (the 'weekend effect') has for many years been attributed to deficiencies in quality of hospital care, often assumed to be due to suboptimal senior medical staffing at weekends. This protocol describes a case note review to determine whether there are differences in care quality for emergency admissions (EAs) to hospital at weekends compared with weekdays, and whether the difference has reduced over time as health policies have changed to promote 7-day services.

**Methods and analysis** Cross-sectional two-epoch case record review of 20 acute hospital Trusts in England. Anonymised case records of 4000 EAs to hospital, 2000 at weekends and 2000 on weekdays, covering two epochs (financial years 2012–2013 and 2016–2017). Admissions will be randomly selected across the whole of each epoch from Trust electronic patient records. Following training, structured implicit case reviews will be conducted by consultants or senior registrars (senior residents) in acute medical specialities (60 case records per reviewer), and limited to the first 7 days following hospital admission. The co-primary outcomes are the weekend:weekday admission ratio of errors per case record, and a global assessment of care quality on a Likert scale. Error rates will be analysed using mixed effects logistic regression models, and care quality using ordinal regression methods. Secondary outcomes include error typology, error-related adverse events and any correlation between error rates and staffing. The data will also be used to inform a parallel health economics analysis.

**Ethics and dissemination** The project has received ethics approval from the South West Wales Research Ethics Committee (REC): reference 13/WA/0372. Informed consent is not required for accessing anonymised patient case records from which patient identifiers had been removed. The findings will be disseminated through peer-reviewed publications in high-quality journals and through local High-intensity Specialist-Led Acute Care (HiSLAC) leads at the 121 hospitals that make up the HiSLAC Collaborative.

## Strengths and limitations of this study

► Difference-in-difference analysis minimises bias from variations in case-mix or institutional characteristics, and allows differentiation of secular trends from the effects of the intervention (specialist intensity).

► Structured implicit case record review encourages a standardised approach to assessment, while permitting reviewers to exercise expert clinical judgement.

► Interassessor variation will be measured through duplicate review of 800 case records, and minimised through standardised training.

## INTRODUCTION

The mortality associated with weekend admission to hospital (the 'weekend effect') has for many years been attributed to deficiencies in quality of hospital care, usually linked to perceptions that senior medical staffing at weekends was suboptimal.[1] The weekend effect has been used as a justification for introducing 7-day services in England.[2] Seven-day services require hospital Trusts in England to meet particular standards for increasing the intensity of weekend senior medical staffing.[3] However, there is little evidence that deficiencies in medical staffing cause the weekend effect. Preliminary research by the High-intensity Specialist-Led Acute Care (HiSLAC) collaboration (www.hislac.org) did not identify an association between weekend–weekday specialist intensity differences and weekend–weekday admission mortality rate differences across the English NHS.[4] However, this was a cross-sectional survey in which no hospital Trust achieved parity between weekend and weekday specialist staffing intensity.

Other potential causes of the weekend effect might include patient factors (severity of illness and case-mix including imperfect case-mix adjustment), contextual factors (resource allocation, health policy, secular trends) and care processes (quality of care). Several studies have identified greater severity of illness among weekend admissions,[5–9] while another reports increased mortality associated with weekend discharge[10]: these suggest a potential community contribution to the weekend effect. Evidence that quality of care in hospital might be worse at weekends comes from an analysis of an Australian regional voluntary critical incident reporting system,[11] but this did not take into account the potential for severity of illness to enhance the opportunity for error, and did not characterise the nature of the critical incidents.

If suboptimal consultant ('specialist') staffing were indeed a cause for the weekend effect, as implied by the 7-day standards policy initiative, there are several ways this might be revealed. First, healthcare error rates might be higher among patients admitted at weekends; second, error typology would include those most likely to be mitigated by the presence of senior physicians (diagnostic accuracy, treatment specificity and timeliness of care); third, a positive association might be expected between the difference in weekend:weekday error rates and the difference in weekend:weekday specialist intensities, indicating that at weekends less reliable care is associated with reduced senior physician presence and fourth, that weekend performance and staffing indices improve with the introduction of 7-day services focused on maximising specialist staffing.

We describe here a protocol for comparing quality of care given to patients admitted as emergencies to 20 hospital Trusts at weekends and on weekdays during two epochs, representing periods before (2012–2013) and during (2016–2017) the implementation of NHS England's 7-day services standards. The outputs from this protocol will be integrated with parallel HiSLAC research workstreams in specialist intensity,[4] a systematic review and framework synthesis,[12] ethnography[13] and health economics.[14]

### Aims and objectives

Using retrospective review of case records from 20 hospital Trusts during two time epochs, this study aims to determine whether there is a difference in quality of care offered to patients undergoing emergency admission (EA) to hospital at weekends compared with weekdays.

The main objectives are as follows:
1. To compare rates of errors and differences in care quality between weekend and weekday admissions,
2. To examine prevalent error types for weekend and weekday admissions,
3. If a difference in error rates or care quality between weekend and weekday admissions is found, examine whether the difference has changed between the two epochs,

4. To inform the Bayesian model proposed in our parallel health economics model.[14]

The co-primary outcomes are the weekend:weekday error rate ratio and the global assessment of care quality. The error rate is calculated as the number of errors per case record from admission to discharge or to 7 days, whichever occurs first.

Secondary outcomes include a comparison of weekend–weekday admissions in error typology and error-related adverse events (AEs), of error rates by day of the week within each admission group and correlation of weekend:weekday error rate ratios with weekend:weekday differences in specialist hours per 10 EAs.

## METHODS AND ANALYSIS

We will conduct a cross-sectional two-epoch comparison of care quality received by emergency patients admitted to hospital at weekends and on weekdays, employing a difference-in-difference analysis[15] to minimise bias from between-Trust differences in case mix and staffing. Conceptually, 7-day services is the research domain, and 'HiSLAC implementation' the research intervention, manifest as an increase in specialist staffing at weekends.

### Participating hospital trusts

From the 121 acute hospital Trusts participating in HiSLAC, 20 will be invited to participate based on Trust size and Sunday specialist intensity derived from the HiSLAC annual point prevalence survey.[4] Trusts will first be grouped into size quintiles based on the number of beds. Within each quintile, hospitals will be ranked by Sunday specialist intensity (hours per 10 EAs) and two Trusts selected from the top and bottom of the intensity spectrum. In the largest Trust quintile, we will also match on number of acute hospitals within each Trust to ensure balance between single-site and multiple-site Trusts.

### Reviewer recruitment and training

We will contact all participating Trusts in England to invite a total of 80 consultants or registrars (residents) in year 5 or above of their training, in an acute adult medical specialty, to act as case record reviewers. Registrar reviewers will conduct their reviews under the indirect supervision of each Trust's HiSLAC local project lead (a senior consultant (attending)). The reviewers will undertake a training course covering the principles of case reviews, providing familiarisation with the scanned case records and permitting practice with samples of case records allowing feedback and discussion.

### Patient case record selection

We will evaluate 4000 non-operative EAs to 20 Trusts to measure variations in quality of care between weekends and weekdays, between high-intensity and low-intensity Trusts, and over time. Case records will be randomly selected from across the whole period of two epochs, financial year 1 April 2012–31 March 2013 and

1 April 2016–31 March 2017, to capture the prepolicy and postpolicy intervention periods of specialist staffing and to avoid bias from seasonal variation. Each epoch will include 100 case records of adult (age ≥17 years) emergency non-operative admissions from each of 20 hospitals, 10 with high-intensity and 10 with low-intensity specialist-led care (2000 case records for each epoch, 4000 case records in total). Half the records from each centre will be weekend admissions (midnight Friday to midnight Sunday) and the other half will be weekday admissions. We have chosen non-operative admissions because the surgical population is currently the subject of a separate national multifaceted quality improvement programme, EPOCH (http://www.epochtrial.org/). The analysis is not restricted to mortality reviews, as it is important to avoid endogenous selection bias (from the outcome influencing the sample). Moreover, deaths are not representative of the general inpatient population,[16] and avoidable deaths formed only 3.6% of 3400 records in a recent review[17] (less than 1% when adjusting for inconsistency between reviewers[18]).

### Case record processing

Patient records will be selected randomly at the University of Birmingham (UoB) from Patient Administration System patient records, which will have been anonymised at each collaborating Trust, allocated an encrypted code and transmitted by secure file transfer protocol to be stored on a secure server at UoB. As case records may be missing, Trusts will be provided with 260 codes of which 65 will be weekend and 65 weekday admissions for each epoch; record retrieval will stop once 50 have been obtained in each group (200 total). Following case selection, the encrypted code will be sent to the Trust medical records department, converted to the local NHS number and the records retrieved for masking to remove patient identifiers (figure 1).

Only the first 7 days following admission will be copied, as the purpose of the review is to focus on care related to the admission phase, not the totality of the patient's course in hospital. A window of 7 days will capture most of the medical care targeted at the acute disease process precipitating admission, and associated AEs linked to the admission. The redacted records will then be scanned as pdf documents, and transmitted by secure file transfer protocol to UoB (figure 1). Thereby, case records will be anonymised and digitised before being presented to the reviewers.

### The review process

We will employ retrospective structured implicit case record review (SICRR) to detect errors in care,[17 19 20] inviting reviewers to produce short narrative judgements with a 5-point Likert scale rating of overall care quality. Reviewers will be asked to identify the location and date of any errors, associate them with specific domains of care (eg, investigation, diagnosis and monitoring), and to make a judgement about whether the errors caused harm to the patient in terms of AEs (figure 2 and online supplementary appendix).

Four thousand case records will be reviewed (100 from each epoch, 200 total from each of 20 trusts). Each Trust's collection of 200 case records will be divided into three (non-overlapping) sets of 80 records as follows: Set I is a random collection of 80 case records from Epoch 1, Set II is a random collection of 80 case records from Epoch

## Steps 1-12 in case record identification and processing

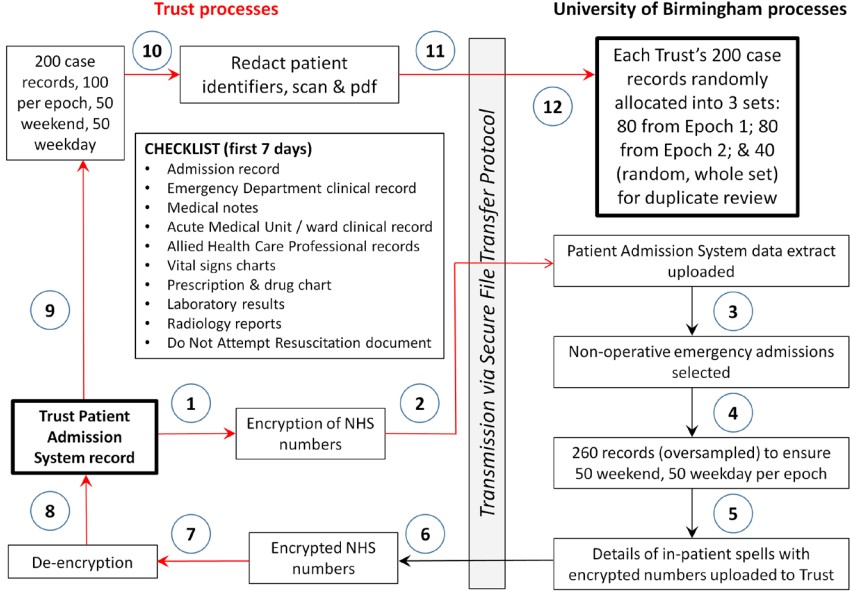

**Figure 1** Case record processing. NHS, National Health Service; PAS, Patient Administration System; UoB, University of Birmingham.

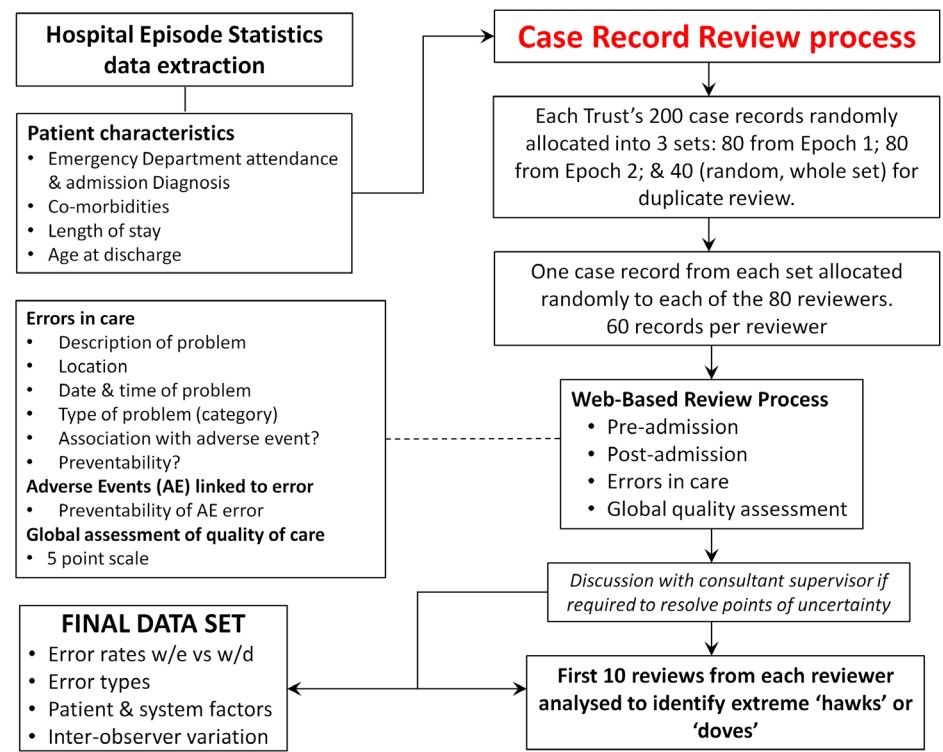

**Figure 2** The review process. AE, adverse event; w/e, weekend; w/d, weekday.

2 and Set III is the remaining 40 case records (20 from each epoch), with each case record duplicated to create 80 records. Each of the 80 reviewers will be randomly allocated just one case record from each of the three sets above, that is, three case records from each Trust. Under this scheme, each reviewer performs 60 reviews and each Trust contributes 40 case records to be reviewed two times; inter-rater reliability can be assessed from the 800 (20%) case notes with independent reviews from different reviewers using the global ratings of care quality.

### Data extraction
Reviewers will access case records online. The case notes will be presented in random order so that learning and fatigue cannot bias the results. Reviewers will not be informed whether records relate to weekend or weekday admissions, but dates will not be masked from the copied records because of the need to determine the timing of events. Reviewers will work independently; they may discuss interpretation of case records with the senior clinician when there is uncertainty, but will be asked to refrain from discussing case records with each other: discussion between reviewers does not improve reliability.[21] The large number of reviewers contributing to each epoch will improve 'calibration', that is, reduce the effect of any outliers ('hawks' and 'doves'). After completion of 10 reviews, we will compare the number of errors identified and global ratings from each of the 80 reviewers to see if there are any extreme hawks or doves.

Data extraction will be performed by the reviewers using a standardised web-based reporting template

(online supplementary appendix 1, based on prior work by Hogan et al[17 22 23] and on the national mortality review process[20]). Assessment of quality of care will be censored at 7 days following admission because the primary aim of the analysis is to examine factors around the time of admission, not later in the patient's stay. Reviewers will be asked to note the date and time (from which elapsed time from admission can be calculated) of any identified errors, so that proximity to the admission point can be recorded. Reviewers will note only those AEs associated with errors, and assess preventability.

### Process and outcome metrics
#### Errors of omission or commission
Errors of omission or commission will be defined as 'the failure of a planned action to be completed as intended or use of a wrong, inappropriate or incorrect plan to achieve an aim' (WHO's taxonomy[24]). Reviewers will be asked to categorise the errors they identify, permitting subsequent comparisons of typology of error (online supplementary appendix 2) in relation to context (weekends, weekdays, night vs day).

#### Adverse events
Adverse events are defined as 'an event that results in unintended harm to the patient by an act of commission or omission rather than by the underlying disease or condition of the patient'.[24 25] Reviewers will grade the preventability of the AE using a standard 6-point scale.[26] This approach permits assessment of the consequences of the errors, without asking reviewers to collect information

on all AEs, many of which will have arisen as a recognised complication or consequence of illness rather than from a lapse in clinical care.

### Global (implicit) measure of quality

Reviewers will provide an overall assessment of care quality, using a 5-point Likert scale in response to the question 'to what extent did this patient receive best-practice care?'

### Specialist intensity ('HiSLAC implementation')

Specialist (consultant) direct involvement in patient care will be derived from the annual HiSLAC point prevalence survey,[4] and expressed as specialist hours per 10 EAs. This self-report web-based survey is administered in June each year; all specialists in 121 participating Trusts across England are invited to record whether they were in the hospital on a particular Sunday or the following Wednesday providing direct clinical care to patients admitted as emergencies, and if so, how many hours they spent doing so.

### Analytical plan

#### Quality of care

##### Error

The primary metric is the number of errors per case record (ie, per episode of care) from admission to discharge or to 7 days, whichever occurs first. The weekend:weekday admission error rate ratio will determine whether weekend admission is associated with more errors in care than weekday admission. We will also examine whether weekend–weekday error rate differences and global assessments of care quality vary by admission epoch and degree of HiSLAC implementation (specialist hours per 10 EAs). We will look for a difference in the difference between weekdays and weekends across low-intensity and high-intensity hospitals, and a further difference between epochs. In this way, each hospital acts as its own control, thereby adjusting for variation in institutional case-mix. We hypothesise that Epoch 1 error rates (the proportion of case records with one or more errors) will be in the region of 20%, and that this rate will diminish between epochs. We will also perform secondary analyses to determine whether errors are more frequent at weekends than weekdays.

#### Adverse event

Rates will be analysed similarly. AEs have been reported consistently to affect around 10% of patient episodes, of which half are reported as being preventable.[26] As we will only document AEs considered to be associated with an error in care, the incidence of AEs in our data set is likely to be lower than 10%, but the preventability rate is higher.

#### Error typology

Error types will be classified by the reviewers using the coding in online supplementary appendix 2. In addition, we will perform a qualitative comparison of the difference between reviewers in their typology of errors. Reviewers' free-text descriptions of error events will be imported into NVivo for further comparative analysis based on a coding frame developed from a sample of transcripts reviewed and discussed by the members of the project team (including clinical and non-clinical members). The focused coding frame will then be applied to the free-text descriptors to derive a separate classification of error types, permitting a comparison between the a priori classification (derived from Hogan et al[17]) and a grounded approach.

#### Health economics

We will use these data to inform the development of the Bayesian model proposed in our parallel health economics modelling.[14] The parameter estimates obtained for error rates from the difference-in-difference in difference type of analysis will be used to update a prior probability distribution obtained from experts. The elicitation of these prior probabilities will be informed by data from the HiSLAC mixed-methods systematic review of the weekend effect[12] and information derived from the parallel HiSLAC ethnographic study of the 20 hospitals.[13] The methodology for elicitation of this 'prior' and for the updating process has been described elsewhere.[14] To calculate the cost utility/benefit of the 7-day services policy of increasing specialist intensity, three further parameters are needed: (a) the error rates must be converted into AE rates and hence into Quality-Adjusted Life Years (QALYs) using the method of Yao et al[27] and Lilford et al[28]; (b) potential cost savings from (any) reduction in AEs will be calculated, again according to Yao et al[27] and (c) the cost of the intervention itself will be calculated building on Meacock et al.[29] Net costs will be derived by subtracting cost (b) from cost (c). The intervention will dominate if these are negative (since such a scenario can only arise if there is a substantial decrease in AEs).

### Statistical analysis

Rates of errors and AEs will be analysed using mixed effects logistic regression models; and the quality of care Likert scale analysed using mixed effects ordinal logistic models, supplemented by linear analyses of the ordinal values. The models will incorporate additive random effects for case notes, reviewers and hospital trusts and fixed effects for day of week (weekend/weekday), time epoch and their interaction. Trust-level weekend effects will be extracted from these analyses. Changes in trust-level effects between epochs will be correlated with contemporaneous changes in specialist involvement estimated from the point prevalence survey.

Reviewer reliability coefficients will be computed as the proportion of the total variance due to variation between case notes.

### Power and detectable differences

The balanced nature of the design ensures that reviewer and Trust effects are eliminated from the comparisons of interest, at least to a good approximation. On the basis of

earlier work,[18] we may assume that 30% of the remaining variance will be attributable to variation between case notes. From the study by Hutchison *et al*,[19] the base error rate is taken as 20% and SD of the Likert global quality scale estimated to be 1.06 (this value is adjusted to take account of the smaller number of Likert categories used in this study, ie, 5 instead of 6).

Using a P value of 5%, the following effects will be detectable with 80% power: (1) a difference in weekend/weekday error rates of 3.2% (from 20 trusts, both epochs), (2) a change over time in the weekend/weekday difference of 6.4%, (3) an average difference between weekend and weekday Likert care scores of 0.08 (which may be interpreted as a shift of one Likert category for 1 in every 12 cases) and (4) a change over time in this difference of 0.17.

## DISCUSSION

This protocol describes using retrospective case record review to determine whether the quality of care received by patients admitted to hospital as emergencies, varies between weekdays and weekends. The study will contribute to our understanding of the 'weekend effect' and the implementation of 7-day services policy by integrating data on error rates and AEs with parallel workstreams on specialist staffing,[4] a systematic review and framework synthesis of the weekend effect,[12] ethnographic evaluations[13] and Bayesian health economics modelling.[14]

The strengths of this novel approach lie in triangulating multiple sources of information on the implementation of national health policy, and the use of difference-in-difference analyses to minimise bias from variations in institutional structures and case-mix.

Case record review is an imperfect instrument for measuring quality of care. Conventional approaches include explicit criterion-based methods (framed by a checklist) or implicit judgement-based methods. While both are widely used, there are important differences between them. Criterion-based review generally provides a higher level of inter-rater agreement than implicit review, but has been criticised for its insensitivity to detecting more nuanced aspects of care quality, and because the criteria themselves may be susceptible to selection bias. The implicit approach allows reviewers to exercise expert clinical judgement in identifying determinants of quality which may have been omitted from predefined criteria, such as evaluating accuracy of diagnosis and appropriateness of care pathways, but this requires a certain level of clinical experience. Midway between these two approaches is the SICRR,[17 19 20] which is the method we will adopt here. Recommended for the national mortality review process supported by the Royal College of Physicians,[20] SICRR invites reviewers to produce short narrative judgements of care quality in specific phases of care, with a 6-point rating of care quality. The case record review process follows that adopted by Benning *et al*,[30] using explicit (criterion based) and implicit (holistic) approaches

since they identify a different spectrum of errors.[31 32] Implicit review is essential to this study since specialist care is most likely to impact on selecting the correct clinical pathway through accurate diagnosis rather than adhering to that pathway once identified, which is where explicit review has its focus.

Our primary measure of quality will be error rates. We have chosen to focus on processes of care rather than outcomes (AEs) because process assessment provides a broader foundation on which to assess safety and quality of care, whereas AEs may arise from the natural history of the disease.[16 31–33] We recognise that reliability between reviewers may be higher for diseases with a strong evidence base[34] and weaker for the likely diverse set of conditions associated with EA to hospital as in this review. To mitigate this effect, reviewers will be asked to categorise the errors they identify, permitting subsequent comparisons of typology of error (online supplementary appendix 2).

Reviewers will also be asked to document whether the errors they identify were associated with an AE, thereby linking process to outcome,[35] and to grade the preventability of the AE using a standard 6-point scale.[27] This permits assessment of the consequences of the errors, without asking reviewers to collect information on all AEs regardless of causation.

We will not restrict the analysis to mortality reviews, because although AEs and preventable AEs are two times as frequent among hospital non-survivors as survivors, and error rates might therefore also be over-represented in this population, these are not representative of the general inpatient population.[16] Random selection of case records from all non-operative admissions will minimise risk of bias which might be associated with random selection of deaths, as the intervention may change the risk of the outcome which would also change the chance of being in the sample (endogenous selection).

### Ethics and dissemination

Informed consent was not required for accessing anonymised patient case records from which patient identifiers had been removed. The findings of this study will be published in peer-reviewed journals, the outputs from this research will also form part of the project report to the Health Service and Delivery Research Programme Board.

**Author affiliations**
[1]University Department of Anaesthesia & Critical Care, University of Birmingham, Birmingham, UK
[2]Institute of Applied Health Research, University of Birmingham, Birmingham, UK
[3]Critical Care Unit, University Hospitals Birmingham NHS FT, Birmingham, UK
[4]NHS Improvement, London, UK
[5]Renal Unit, Heart of England NHS Foundation Trust, Birmingham, UK
[6]Southern Health NHS Foundation Trust, Southampton, UK
[7]University Hospital Southampton NHS Foundation Trust, Southampton, UK
[8]Department of Health Sciences, University of Leicester, Leicester, UK
[9]Member of the Academy of Medical Royal Colleges Patient Liaison Group, London, UK
[10]Health Services Management Centre, University of Birmingham, Birmingham, UK
[11]Division of Health Sciences, University of Warwick, Coventry, UK

**Contributors** JB and RL developed the initial study idea. CPA, JB and AB obtained statutory and ethics approval. JB, CPA, AG, GR, CB, TE, RMT, CR, MC, AB, CT, ES, JS, PR, RM, Y-FC, SIW and RL informed the protocol methodology. All authors contributed to writing the final report and approved the final version.

**Funding** This project was funded by the National Institute for Health Research, Health Service and Delivery Research Programme (project number 12/128/17).

**Disclaimer** The views and opinions expressed therein are those of the authors and do not necessarily reflect those of the HS and DR, NIHR, NHS or the Department of Health.

**Competing interests** JB is the Chair, NICE Acute Medical Emergencies Guideline Development Group National Clinical Guideline Centre.

**Ethics approval** The HiSLAC study, including the case record reviews, was approved by the Sub-committee of the South West Wales ResearchEthics Committee, reference13/WA/0372, on 11 November 2013.

**Provenance and peer review** Not commissioned; externally peer reviewed.

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
