## [Reviewer comments · BMJ Open]

ARTICLE DETAILS

TITLE (PROVISIONAL)	A Two-Epoch Cross-Sectional Case Record Review Protocol Comparing Quality Of Care Of Hospital Emergency Admissions At Weekends Versus Weekdays
AUTHORS	Bion, Julian; Aldridge, Cassie; Girling, Alan; Rudge, Gavin; Beet, Chris; Evans, Tim; Temple, R.; Roseveare, Chris; Clancy, Mike; Boyal, Amunpreet; Tarrant, Carolyn; Sutton, Elizabeth; Sun, Jianxia; Rees, Peter; Mannion, Russell; Chen, Yen-Fu; Watson, Samuel; Lilford, Richard

VERSION 1 – REVIEW

REVIEWER	N Arulkumaran Bloomsbury Institute for Intensive Care Medicine. University College London Division of Medicine 235 Euston Rd Bloomsbury London NW1 2BU UK
REVIEW RETURNED	27-Aug-2017

GENERAL COMMENTS	This is a novel and clinically relevant study proposed by experts in the field There are a number of strengths to the study- particularly focusing on the errors rather than mortality. The findings of this study will have a direct implication on clinical workforce planning. There is a significant body of work that has been completed or ongoing as part of this research collaborative I am unable / not suitably qualified to comment on the statistical analysis of qualitative studies There are a few points which the authors may want to discuss or consider: • I presume the rationale in following patients up for 7 days is to capture any late complications related to errors on admission. However, there will be errors that occur at any point during the 7 day period. At which point will the error no longer be considered to be associated with the acute admission? Will it be within the first 24hrs, until seen by a specialist, or at any time till day 7?• Is there any value in grading adverse events (e.g. near misses, serious untoward events, and never events)?• Will inter-observer variability be assessed as part of training?
---

	 • The investigators suggest that there may have been an increase in specialist staffing since the introduction of seven day services. There may be an interaction between the change in intensity of specialist staffing at each unit over the 2 epochs (low to high, low to low, high to high intensity). Will this interaction be evaluated? • The study inclusion criteria is patients admitted to the hospital. A small but significant proportion of patients may have presented to the A+E but discharged home; only to re-present and get admitted hours to days later. Will it be possible to capture the cohort of patients who should have been admitted the first time round? • Will efforts be made to ensure that trainees/ registrars avoid assessing case notes from the trust they work at? Assessing case notes where they know the medical team, or where they might work themselves, may bias the results (I appreciate the investigators will be assessing inter-observer variability, but it would be best to try and avoid bias before they get to this stage) • Hospitals will be divided into having either high or low intensity specialist staffing based on the HiSLAC definition, which is based on staffing levels on Sunday or Wednesday. Whilst I understand that the objective is to assess the level of staffing in relation to errors, it may useful to define if errors were made before or after specialist review. Majority of the errors may be made prior to specialist review, irrespective of the day of week. • The assumption is that consultant error rates do not vary by day of week. Would it be possible and valuable to ascertain if the errors were made by a junior trainee, senior trainee, or consultant? • Another time when specialist intensity is lower is overnight. Will there be a subgroup analysis or secondary analysis of errors made overnight? There may be an interaction between intensity of specialist staffing, time of day, and errors • The complexity of patients may vary by day of week and the errors may increase with the complexity of the patient (Concha OP, et al. Do variations in hospital mortality patterns after weekend admission reflect reduced quality of care or different patient cohorts? A population-based study. BMJ quality & safety. 2014;23(3):215-222). Is there any way to correct for this? • What are the timelines for getting this study complete?
--	--

REVIEWER	Michael Gillies University of Edinburgh, Edinburgh, UK. No Competing Interest
REVIEW RETURNED	03-Sep-2017

GENERAL COMMENTS	Thank you for the opportunity to review this protocol paper for a casenote review study to determine the "weekend effect" on the quality of care delivered in NHS trusts in England. This area is topical and highly relevant to healthcare planning and resource allocation. It is an area of controversy and although there are now many studies investigating the weekend effect, quality is
---

	variable and results conflicting. The authors are to be commended for their robust approach to this perplexing area of health services research. This research team are recognised experts with a track record of delivering research in this field. The protocol paper is well written. The study is complex and so my understanding would benefit from clarity on the following points:  1. Primary and secondary objectives and outcomes could be stated more clearly, rather than "main objectives" and "secondary objectives" (pp6 Line 5-16). One assumes that the primary outcome is the "error rate" as defined on page 8 lines 52-56? Also for "specialist intensity" (an outcome metric or the intervention?)...is this defined as "specialist hours per 10 emergency admissions"? This is implied elsewhere in the manuscript, but not stated in the Process and Outcome Metrics definitions section pp9 lines 18-25. 2. The "research intervention" is stated as an increase in specialist staffing at the weekend or HiSLAC implementation. Will this study report if this has been achieved and how successfully seven day services have been implemented or has this been published elsewhere? If not, is this a research objective? Clearly it would be useful to know if the research intervention has been successfully implemented between the two epochs. 3. How will missing or incomplete data in the Patient Case Records be dealt with? Inadequate documentation could be a marker of poor care itself. 4. Will the reviewers require a minimum number of years in training or post qualification? Specialist trainees can be of wide variation in experience and will be commenting on consultant led care in many instances. 5. Would the authors expect confounding arising from current standards of care being applied to historic cases by reviewers and if so how can this be minimised? 6. In "Statistical Analyses" the authors state that the rate of outcome of interest (error rate) will be "analysed using logistic regression analysis". The models will adjust for differences in NHS trusts and reviewers, but will there be any adjustment for patient level factors or case mix e.g. Admission diagnosis, age, comorbidity, deprivation etc. ? Thanks for the opportunity to review this ambitious study, I look forward to reading the results when complete.
--	---

REVIEWER	David Metcalfe University of Oxford, U.K. I hold an honorary contract with the University of Warwick, which is where a number of these authors are based. I do not believe that I have met or worked with any of the authors, except for Chris Beet with whom I worked briefly in 2013. I am a junior doctor and former
-----------------	--

	member of the British Medical Association.
REVIEW RETURNED	06-Sep-2017

GENERAL COMMENTS	This is clearly an important project and the authors have taken considerable care to approach the question in a methodologically rigorous manner. Despite the efforts to blind case note reviewers, the group of patients admitted at weekends will still be readily identifiable. There is considerable risk of bias given the political context of the research question, the subjective nature of identifying adverse events, and the fact that reviewers are unavoidably participants in the "weekend effect" controversy. This is mitigated by the collection of more objective process measures, e.g. time to consultant review, etc. I hope that the objective measures will also be presented and given a central place in the final manuscript. I am also not completely convinced that adverse events will be reliably identified by case note reviews. Anecdotally, I suspect that many errors (particularly those that do not lead to harm) go undocumented. We know that coding accuracy varies between weekends and week days (BMJ 2016;353:i2648), and it is feasible that documentation completeness will vary as well. It will be open to interpretation whether or not differences in documentation completeness at weekends truly reflect differences in quality of care. I have recommended publication of the protocol without any need for changes.
---

VERSION 1 – AUTHOR RESPONSE

Reviewer #1:

* We appreciate this question which has allowed us to provide more detail about our proposed methodology. The answer to the reviewer's question is 'any time out to day 7'. Our (co-) primary metric is the number of errors per case record over the whole of the patient's stay up to seven days. We have clarified this in the first paragraph of the Analytical Plan (page 7). By comparing weekend admissions with weekdays, this analysis answers the question 'Is admission to hospital at weekends associated with more errors in care than admission on weekdays?' In other words, this is a comparison of the episode of care (to day 7) based on the admission type (weekend/weekday admission), not a comparison of the admission time period (weekend vs weekday).

We will also perform secondary analyses comparing the time period, to determine whether errors are more frequent at weekends than weekdays (directed at the slightly different question 'Are weekends riskier than weekdays?').

We have chosen 7 days as representing that period of time when medical care will be targeted most particularly at the acute disease process responsible for precipitating hospital admission. It will also, as the reviewer proposes, provide a sufficient period to identify adverse events associated with the admission (revised text in 'Case Record Processing, second paragraph, page 5). We chose not to study the whole hospital stay given the highly skewed distribution – analysis of HERS data shows the median length of stay of emergency admissions to the NHS is one day.

* We considered this option, and felt that rather than asking reviewers to try to assess the nature and severity of AEs, preventability would be more useful in determining the nature of errors and their consequences. We are only recording those AEs considered to be attributable to an error in care, and limiting the enquiry to seven days, which means that longer term outcomes (hospital, community) will not be available to reviewers and this would limit their ability to assess severity of AEs.

* We will assess inter-observer variability in the project, but not in the training sessions as we do not have the time or technology available to undertake this analysis. We have three training days planned, during which our main focus is for the reviewers to learn to use the on-line system and to develop a harmonized approach to identifying and grading error and adverse events. Page 6, 3rd paragraph: ; 'inter-rater reliability can be assessed from the 800 (20%) case-notes with independent reviews from different reviewers using the global ratings of care quality' Page 9, second para, 'reviewer reliability coefficients...'

* We have already categorized hospitals by baseline specialist intensity on a Sunday, using the data from our annual survey of Sunday:Wednesday differences. We can therefore measure whether there have been changes since 2014. We will examine direction of trends in the manner suggested, as secondary analyses, but given the comparatively small number (20) of hospitals in this evaluation we may not pick up a signal.

* We will not be able to analyse attendances preceding the index admission as case records will only be copied from the index admission entry, consonant with most research into the weekend effect. We will however try to capture pre-admission community care (eg: source of referral, mode of presentation).

* As we are randomly assigning case records to reviewers, and as reviewers are drawn from across the NHS and will have worked in several different hospitals at different times, it would be difficult to prevent the possibility that some reviewers may receive case records from hospitals with which they are familiar. However, the likelihood of this happening will be randomly distributed, as will potential biases. No reviewer will receive more than 3 case records from the same hospital, which will also minimize risk of bias (3rd paragraph page 6).

* We will ask reviewers to document the time and date that errors occurred, if they are able to do so (this is an imprecise science). We are also asking them to note the time and date of first consultant review – again, if they are able to do so. If the data are sufficiently secure, we will determine the proportion of errors made before, and after, consultant review.

* We have not assumed that consultant error rates are a constant, but we have assumed that when consultant input occurs, the standard of care provided will be that expected of a consultant. It would be of interest to see if seniority is associated with different types of error, but it will be very difficult to do this securely given the difficulty of case record review – legibility, level of documentation and so on.

* We will note the time of day that errors occurred, and could provide this as a secondary analysis. However, it is difficult to compensate for differences in opportunity for error using case record reviews (eg: lower activity at night, lower opportunity for error). We will take the opportunity to see if there are differences in error typology by day of week or time of day (see Process and Outcome Metrics, first para, page 7).

* We agree with this possibility. Comparing weekends with weekdays within the same institution will tend to minimise case mix differences, but will not completely address the possibility that weekend

admissions may be sicker (or that fewer low-risk patients are admitted at weekends). The HES dataset allows us to adjust for baseline differences in diagnosis and chronic disease, but not in acute physiological disturbance (currently a separate project for our group – we have found only a very modest contribution to the weekend effect from physiology). We will therefore examine error-rates by day of week within each admission group. This may provide some insight into whether any weekend effect is patient- or context- dependent (error rates by day of week added to secondary outcomes, Page 4, fourth paragraph).

* Subject to recruiting sufficient reviewers, the review process will start in February and end in June/July 2018. We expect to have completed the analyses before the end of 2018.

Reviewer #2:

* We have added a paragraph at the end of 'Aims and Objectives' describing the primary and secondary outcomes: "The co-primary outcomes are the weekend-weekday error rate ratio and the global assessment of care quality. The error rate is calculated as the number of errors per case record from admission to discharge or to seven days, whichever occurs first. Secondary outcomes include a comparison of weekend-weekday admissions in error typology and error-related adverse events; of error rates by day of the week within each admission group; and correlation of weekend:weekday error rate ratios with weekend:weekday differences in specialist hours per 10 emergency admissions (EAs)." (Page 4).

Apologies for omitting the definition of specialist intensity – thank you for noting this. We have added the definition to the final paragraph in 'Process and Outcome Metrics'. Specialist intensity is indeed a key component of the intervention (Page 7).

* We are using the putative increase in specialist staffing (specialist intensity = hours/10 emergency admissions) as the key indicator that seven day service standards are being implemented. We will report changes in specialist intensity for these 20 hospitals between the two epochs and explore associations between staffing and error rates (see secondary objectives, page 4). However, our main objective is to see if we can detect a difference in error rates between weekend admissions and weekday admissions, as a potential explanation for the weekend effect.

* We offer reviewers the option to record 'unable to determine' in response to a number of areas of enquiry (Appendix). We will take this to indicate inadequate documentation. We cannot however translate this directly into 'error' unless the reviewer explicitly identifies documentation failure in that way. We will cover this issues in reviewer training.

* Reviewers are specialist registrars in year 5 or above, or consultants, in an acute adult medical speciality (emergency medicine, acute medicine, internal medicine participating in the acute take, or intensive care medicine (including specialities paired with intensive care)). We have added this explanation to the section 'Reviewer Recruitment and Training (first paragraph, page 4).

* Best practice guidance may indeed have changed between the two epochs. Had we proposed to use criterion-referenced reviews we would have had to apply best practice standards appropriate for the epoch in question for a large number of disease processes. This would have been far too complex to implement. We have therefore chosen structured implicit review. While this does indeed run the small risk that reviewers might apply 2016-17 standards to 2012-13 care processes, given the wide range of possible errors, we believe that those relating to specific treatment pathways will form a relatively small proportion.

* The sample size (4000 patients) is too small to make judgements about the impact of case mix on error rates and patient outcomes. We will record patient factors such as age, sex, Charlson score, income deprivation component of the Index of Multiple Deprivation 2015, and diagnostic category to confirm that the aggregated case mix for the 20 hospitals is similar to that of the NHS in England, to support generalizability of results.

Reviewer #3:

* This is certainly a possible source of bias. We suspect however that reviewers will have enough work to do in reviewing the case records without explicitly consulting diaries to work out whether a specific admission occurred at a weekend or on a weekday. In the training sessions we will ask reviewers to refrain from this sort of gaming. All the objective measures will be published in the final manuscript.

* Case record review is indeed an imprecise science. Some errors may well go unrecorded. Reviewers will be sensitized to documentation failures. For example if there is no record of timely prescription or timely administration of antimicrobials in sepsis, the reviewer may choose to identify this as a potential error in care.

Interestingly, we have actually found better documentation of vital signs (NEWS score) at weekends than on weekdays in a large single centre study (data on file). The paper to which the reviewer refers (by Professor Rothwell from Oxford) shows 'no bias in distribution of weekend versus weekday admission of the 319 strokes missed by coding', and the coding errors were predominantly administrative in origin, not a failure of documentation in the case records.

VERSION 2 – REVIEW

REVIEWER	N Arulkumaran University College London, UK
REVIEW RETURNED	30-Oct-2017

GENERAL COMMENTS	The authors have given due consideration for all reviewers' comments and have updated the manuscript accordingly. All inherent limitations of the study have been openly addressed and discussed. I look forwards to the results of the study!
--

REVIEWER	Michael Gillies Royal Infirmary of Edinburgh, Little France Crescent, Edinburgh, UK
REVIEW RETURNED	01-Nov-2017

GENERAL COMMENTS	Thank you, the authors have satisfactorily addressed my comments.
---